# Pediatric Narcolepsy—A Practical Review

**DOI:** 10.3390/children9070974

**Published:** 2022-06-29

**Authors:** I-Hang Chung, Wei-Chih Chin, Yu-Shu Huang, Chih-Huan Wang

**Affiliations:** 1Department of Child Psychiatry and Sleep Center, Chang Gung Memorial Hospital and College of Medicine, Taoyuan 333, Taiwan; shadow10558@cgmh.org.tw (I.-H.C.); auaug0327@hotmail.com (W.-C.C.); 2Department of Psychology, Zhejiang Normal University, Jinhua 321004, China; chihhuan1@gmail.com

**Keywords:** pediatric narcolepsy, hypersomnia, daytime sleepiness, cataplexy, neuroimage

## Abstract

Pediatric narcolepsy is a chronic sleep-wakefulness disorder. Its symptoms frequently begin in childhood. This review article examined the literature for research reporting on the effects of treatment of pediatric narcolepsy, as well as proposed etiology and diagnostic tools. Symptoms of pediatric narcolepsy include excessive sleepiness and cataplexy. In addition, rapid-eye-movement-related phenomena such as sleep paralysis, sleep terror, and hypnagogic or hypnapompic hallucinations can also occur. These symptoms impaired children’s function and negatively influenced their social interaction, studying, quality of life, and may further lead to emotional and behavioral problems. Therefore, early diagnosis and intervention are essential for children’s development. Moreover, there are differences in clinical experiences between Asian and Western population. The treatment of pediatric narcolepsy should be comprehensive. In this article, we review pediatric narcolepsy and its treatment approach: medication, behavioral modification, and education/mental support. Pharmacological treatment including some promising newly-developed medication can decrease cataplexy and daytime sleepiness in children with narcolepsy. Other forms of management such as psychosocial interventions involve close cooperation between children, school, family, medical personnel, and can further assist their adjustment.

## 1. Introduction

Pediatric narcolepsy is a rare disease based on the global average prevalence between 20 and 50 per 100,000 [1,2,3,4,5,6,7,8]. It is a chronic sleep-wakefulness disorder. Currently, patients with narcolepsy can be divided into two types based on the presence or absence of cataplexy according to the International Classification of Sleep Disorder—Third Edition (ICSD-3) [9]. The first type of narcolepsy recognized by the ICSD-3 is narcolepsy type 1 (previously narcolepsy with cataplexy). In addition to excessive daytime sleepiness (EDS), patients with narcolepsy type 1 also suffer from cataplexy and/or low cerebral hypocretin levels (Figure 1a). The other type is narcolepsy type 2 (previously narcolepsy without cataplexy). Patients with narcolepsy type 2 do not have cataplexy, and their hypocretin levels are normal or have not undergone cerebral spinal fluid (CSF) examination (Figure 1b). 

Although diagnostic criteria in ICSD-3 outlined the common symptoms and signs of narcolepsy, several differences in presentation have been noted between adult and pediatric narcolepsy patients. In addition, the symptoms of each narcoleptic patient can change between different life stages. For example, some children who develop cataplexy later may be diagnosed as having narcolepsy without cataplexy initially [10]. The differences between children and adult population may be crucial in diagnosis and tailoring treatment strategies for pediatric narcolepsy.

## 2. Symptoms/Signs of Pediatric Narcolepsy

Children with narcolepsy experience constant excessive daytime sleepiness and sudden sleep attacks during activities at any time of the day [9]. Other common symptoms/signs may also occur. We should note that these symptoms might appear gradually during the course of the disease, and not every child experiences all the symptoms.

Excessive daytime sleepiness (EDS): EDS is present in all patients with narcolepsy and often noted as the first symptom/sign in children. Children may complain of mental cloudiness, fatigue, sleepiness, forgetfulness, low energy, and difficulty concentrating, as well as have related behavioral problems, such as irritability, hyperactivity, social withdrawal, depression, or even aggressiveness. EDS impairs children’s functions and disturbs daily activities, including school and social life. Sleep attacks last longer in children than adults; preschool children can still experience tiredness soon after afternoon naps for 2–3 h, and the need for napping can persist into childhood after age 5–6.Cataplexy: Patients with cataplexy experience muscle weakness or a sudden loss of muscle tone, which can be noted in 70% of children with narcolepsy. It is usually triggered by stress or emotion, such as laughter when seeing something funny or being scared. The duration of cataplexy is brief, lasting a few seconds to several minutes, and the severity differs. It can be as mild as simply the feeling of weakness of eyelids or legs, but can also be severe, for example with body paralysis, which can result in falling or injuries. Cataplexy in children has some unique features, including sticking out of the tongue, facial and eyelid weakness, and abnormal facial movements and expressions. It can be wrongly labeled as clumsiness, seizures, or attention-seeking behavior in young children.Sleep paralysis: When patients have sleep paralysis, they find that they cannot move or speak after they just wake up or before they fall asleep. This symptom does not last long, usually seconds to a few minutes.Hallucinations: Hallucinations in patients with narcolepsy occur before sleep (hypnogogic hallucination) or after waking up (hypnopompic hallucination). Patients with pediatric narcolepsy often experience vivid and dream-like events, often involving images or sounds that can be scary.Disturbed sleep through the night: Patients with narcolepsy not only have daytime sleepiness but also disturbed the sleep during nighttime. Although they do not have problems falling asleep, narcoleptic patients often report disrupted nighttime sleep and difficulty maintaining sleep.

In addition to the above typical symptoms, there are other characteristics. First, autonomic behaviors may be noted. Patients fall asleep but continue their ongoing activities, such as writing or other daily tasks. They are not aware and do not remember what they do during autonomic behaviors. Second, obesity is common in patients with narcolepsy. It can be noted early in childhood during the disease course. Up to 25–50% of children with narcolepsy are obese [11,12]. Early onset of puberty related to hypocretin is also pointed out by previous studies. Children with earlier age of onset have a greater risk of precocious puberty [13].

## 3. Prevalence and Etiology

The reported prevalence of narcolepsy varies in studies from different countries and with different ethnicities. The lowest prevalence is reported in Israel (0.23 per 100,000), while the highest is in Japan (160 per 100,000) [14,15]. According to previous reports, the global average is between 20 and 50 per 100,000 [1,2,3,4,5,6,7,8]. Although some studies suggest a higher prevalence in males [14], narcolepsy affects males and females equally.

One current hypothesis is that narcolepsy is related to the destruction of the specific brain area responsible for sleep and wake function and the loss of hypocretin, a neuropeptide in the brain. Accumulating evidence has proven that narcolepsy type 1 (previously narcolepsy with cataplexy) is caused by the loss of hypocretin-1 (orexin) neurons in the lateral hypothalamus [16,17,18], but patients with narcolepsy type 2 are found to have normal hypocretin levels, and its etiology remains unknown [19]. 

Tracing back to 1983, Juji et al. (1984) found that narcolepsy was associated with the human leukocyte antigen, HLA-DR [20]. Both environmental and genetic factors were assumed to play important roles in the pathogenesis of narcolepsy [21]. Previous studies have shown that most patients with narcolepsy type 1 and half of patients with narcolepsy type 2 carried HLA-DQB1*06:02 [9,22]. The genetic susceptibility of narcolepsy can result from the heterodimer formed by HLA-DQA1*01:02 and HLA-DQB1*06:02, an antigen presenter to the T cell receptor (TCR). Polymorphism in non-HLA genes affecting immune regulation can also be connected to narcolepsy. Other studies found that narcolepsy is associated with TCR α polymorphisms and anti-TRIB2 antibodies [23,24]. All these findings support the hypothesis of autoimmune destruction of hypocretin cells in the pathophysiology of narcolepsy. 

Some studies found factors such as prenatal nutrition, obesity, and stress may also affect hypocretin neurons [25,26,27]. Early life environmental factors can have a role in the damage of hypocretin neurons, supported by the seasonal predominance of birth (such as May and June) in patients with narcolepsy [28,29]. Infections have been proposed as a potential trigger for the autoimmune mechanism. In 2009, Aran et al. (2009) reported that streptococcal infections are probably a significant environmental trigger for narcolepsy [30]. Several recent studies have shown increased cases of narcolepsy, especially in children and adolescents, in relation with H1N1 influenza, as described above. Although recent global research does not support the association between the vaccines for H1N1 and narcolepsy [31], an interesting finding in Taiwan is that narcolepsy can be triggered by the H1N1 infection itself [32].

Streptococcus and H1N1 infections may trigger narcolepsy by activating T-cells and B-cells. The first possible mechanism is molecular mimicry [33,34]. Antigen from streptococcus or H1N1 is presented by the antigen-presenting cells (APC) in major histocompatibility complex (MHC)-DQA1*01:02-DQB1*06:02, and T-cells are activated after recognizing the antigen through TCR. Subsequently, T-cells may recognize the hypocretin antigen of hypocretin neurons via cross-reactivity and attack these neurons. This molecular mimicry occurs through B-cells, which directly recognize the antigen, but also requires T-cell activation. Other proposed mechanisms include superantigen activation and stander activation. Superantigens such as streptococcus can link TCR on T-cells with MHC II molecules expressed by cells independent of antigen specificity, and said process leads to TCR signaling and T-cell activation. When neurons express low levels of MHC II molecules, T-cells are unlikely to interact in an antigen-dependent way. However, in stander activation, T-cells can be activated by generalized immune activation, which is also independent of specific antigens [19]. 

## 4. Evaluation and Diagnosis of Pediatric Narcolepsy

The onset of narcolepsy symptoms usually occurs between the ages of 10 and 30 years, generally during the second decade of life. The peak is at 14 to 15 years, although symptoms can be present in children less than age 10 [35,36,37,38]. Levy et al. (2019) reported on 42 narcolepsy children at the Royal Hospital for Children between 1996 and 2016. The time between symptom onset and diagnosis was shorter than that reported with the adult, but diagnoses were still delayed about one year, with the longest delay up to 11 years [39]. 

Delayed diagnosis in pediatric narcolepsy may be caused by a variety of reasons. Related behavioral problems of pediatric narcolepsy are often thought to be psychiatric conditions, while cataplexy can be misdiagnosed as normal falls, epilepsy, or other neurological disorder. During school, it is not uncommon to see children appear drowsy, slouched over their desks, or even fall sleep. These conditions make diagnosing narcolepsy in children difficult [10]. 

Young children with narcolepsy may not be diagnosed in preschool or school stages before adolescence or early adulthood. Daytime sleepiness may not be obvious in younger children and is often missed. Other narcoleptic symptoms, such as hypnagogic and/or hypnopompic hallucinations and sleep paralysis, are not always detectable.

Sleep behavior is often believed by teachers and parents to be apathy, pathological sleepiness, or even normal napping. Prolonged sleep, dreamy sleep, and difficulty waking up are commons, but sleepiness can be masked by such abnormal behavior as irritability, aggressiveness, social withdrawal, or shyness [40,41,42]. Frequent cataplectic attacks at an early age should lead to detailed clinical, neuroimaging, and other brain examinations to rule out a secondary etiology.

To evaluate and diagnose pediatric narcolepsy, the following procedures and tests are recommended and can be used:
To make a narcolepsy diagnosis and rule out other causes of sleep symptoms, it is important to collect a detailed medical history and have a thorough physical examination performed by a pediatrician and sleep specialist. Parents may not detect cataplexy symptoms as well as the children who experience it, but they have better recognition in the circumstances of children’s sleepiness. A detailed sleep history should be gathered from the child, their parents, and their teachers. Some questionnaires, such as the “Pediatric Daytime Sleepiness Scale (PDSS)”, are helpful in collecting information about sleepiness and treatment responses [42]. Other questionnaires include “Cataplexy Diary” and “Epworth Sleepiness Scale-Child Adolescent (ESS-CHAD)”. Cataplexy Diary was modified in its definitions and examples of cataplexy by using child-friendly terminology and adding a quantitative question for frequency and standardization for evening administration with self-completion. ESS-CHAD was modified by child-friendly wording to ensure that items reflect children’s activities and environments. ESS-CHAD comes in two versions: one with a one-month recall period for general use, and the other for research, with a recall period of “since your last study visit” (as short as one week) [43].Actigraphy recordings and sleep log are useful for ruling out other sleep disorders. Actigraphy measures movement and collects data via a wrist-watch type device. It is worn for up to 2 weeks, during which time the child and parents need to fill out a sleep log to document sleep and wake times. Filardi et al. (2016) collected the actigraphy data of 22 drug-naïve type 1 narcolepsy children and 21 age- and sex-matched controls for seven days during the school week. The results showed actigraphic sleep measures have good discriminant capabilities in assessing narcolepsy type 1 nychthemeral disruption in drug-naïve children and indicated the sensibility of actigraphic assessment in the diagnostic work-up of childhood narcolepsy type 1 [44]. Actigraphy also offers the possibility to longitudinally follow up children and has the potential to become a key tool for tailoring treatment in pediatric patients [45,46]. Two standard sleep study tests are traditionally performed. The overnight polysomnography (PSG) and the multiple sleep latency test (MSLT) are still the “gold standards” for diagnosing narcolepsy (Table 1). PSG and MSLT should be preceded by at least one week of actigraphic recording or a sleep log [9].The blood test of HLA DQB1*06:02 haplotype has also been demonstrated to be associated with narcolepsy. HLA is a gene region encoding MHC protein located on chromosome 6 and divided into three sub-regions (class I, II, and III). Studies by Mignot and colleagues have shown that narcolepsy with cataplexy is highly associated with HLA DQA1*01:02 and HLA DQB1*06:02 in all ethnic populations [47]. Other studies have had similar findings [22,48,49]. Another test is measurement of the CSF hypocretin levels (<110 pg/mL). Kanbayashi et al. (2002) showed that from early infancy to old age, the CSF hypocretin level remains stable, and thus a low or undetectable hypocretin level is a very valuable diagnostic marker for pediatric narcolepsy [50]. However, this test is not commonly performed since it requires lumbar puncture [51], an invasive method that raises many concerns for parents. Neuroimages: Neuroimages may provide clinicians another tool for differential diagnosis for pediatric narcolepsy patients [22,49]. Although inconsistent, PET studies of adolescents with narcolepsy type 1 revealed hyper- and hypometabolism in many cortico-frontal and subcortical brain regions, which had significant correlations with neurocognitive test performance [49,52,53,54,55,56]. These PET findings parallel those in structural neuroimaging studies, which showed a reduction of cortical gray matter in the frontotemporal areas [49,56,57,58]. A recent study reported that the subtle structural brain changes involving attentional and limbic circuits could be detected in children and adolescents with narcolepsy type 1 [59]. Another recent study reported that patients with type 2 narcolepsy showed significantly less hypermetabolism in the fusiform gyrus, striatum, hippocampus, thalamus, basal ganglia, and cerebellum than in patients with type 1 narcolepsy via FDG-PET [22].


## 5. Treatment Strategy of Pediatric Narcolepsy

Narcolepsy is a chronic neurological disorder that currently has no cure. The goal of treatment is to reduce daytime sleepiness and other disturbing symptoms and improve daytime function and quality of life, so that children can live as near a normal life as possible. Treatment plans for childhood narcolepsy should be comprehensive and typically involve pharmacological and non-pharmacological approaches. In addition, similar to the previous AASM guideline [60], the European guideline and expert statements on the management of narcolepsy published in 2021 suggest that the treatment of narcolepsy should include both control of daytime sleepiness and other nocturnal sleep related symptoms [61]. Compared to adult narcolepsy patients, whose treatment strategy also included pharmacological and non-pharmacological approach, they had more approved medication for clinical use [61,62]. When prescribing medications for pediatric narcolepsy patients, it is important to acknowledge evident-based safety and the growth issues within each medication. It is also important to provide sufficient psychological counseling and behavioral modification for pediatric narcolepsy patients in multiple environments, and to enhance the cooperation between school teachers, family members, and the medical team.

Pharmacological treatment: Medications are used to treat EDS and other narcoleptic symptoms, including cataplexy, sleep disturbances, and hallucinations. Medications approved by the US Food and Drug Administration (FDA) for pediatric narcolepsy including methylphenidate and amphetamine for EDS and sodium oxybate for EDS and cataplexy. However, only sodium oxybate have been proven effective in pediatric patients through randomized placebo-controlled studies. Recently, guidelines about treatment strategy of narcolepsy were published both in Europe and America. In the European guideline and expert statements on the management of narcolepsy, they recommend sodium oxybate for EDS and cataplexy in pediatric patient [61]. Meanwhile, in the American Academy of Sleep Medicine (AASM) clinical practice guideline of hypersomnolence, besides sodium oxybate, Modafinil is also suggested for pediatric narcolepsy patients [62]. The overview of medication suggested for pediatric narcolepsy by both guidelines was presented in Table 1.

Excessive daytime sleepiness:
(1)Traditional central nervous system stimulants: For pediatric narcolepsy, amphetamine and methylphenidate can be tried. Methylphenidate and amphetamine promote wakefulness by increasing synaptic concentrations of dopamine [63]. Methylphenidate may be started from 5 mg three times per day, and gradually increasing to maximum dose 60 mg/day for ADHD pediatric patients. Common side effects of stimulants include restlessness, nervousness, headache, loss of appetite, and heart discomfort. In children, the starting dose should be low and increased gradually as needed with careful monitoring. High blood pressure, arrhythmia and drug misuse have been reported [64,65]. Although not recommended in AASM guideline, stimulant such as methylphenidate had been used in clinical practice in many countries, especially in Asia.(2)Wake promoting agents: Wake promoting agent such as Modafinil was also suggested for treating EDS in pediatric narcolepsy [62]. Modafinil’s wake promoting mechanism was hypothesized from the increases in monoamines releasing, especially the elevation of hypothalamic histamine levels [66]. Modafinil is usually started from 50 mg/day, and gradually increased to maximum 200 mg/day [67,68]. In the clinical practice, for pediatric narcolepsy patients, we suggest starting from low doses, and gradually titrating medication until symptoms ameliorate. Compared to traditional stimulants, Modafinil has fewer side effects and is less addictive.(3)Other agents acting on γ-Aminobutyric acid (GABA): Sodium oxybate is the only FDA-approved medication to treat both daytime sleepiness and cataplexy in patients age 7 years and older with narcolepsy [69]. Filardi et al. (2018) conducted a long-term study of sodium oxybate in children. They used the actigraphy to record the sleep and wake profile objectively after 1 year sodium oxybate treatment. Their results showed that symptoms severity of narcolepsy and children’s anthropometric features were changed and improved as expected [45]. Although the mechanism of action of sodium oxybate is not fully understood, therapeutic effects on EDS are hypothesized to be mediated through modulation of GABA-B receptors during sleep [70]. Sodium oxybate usually started from 3 g/day, and may gradually increase to maximum nightly dose of 9 g [71,72].
Cataplexy:

Sodium oxybate was an evidence-based and guideline support medication for treating cataplexy in patients aged 7 years and older with narcolepsy [61,69]. 

Antidepressants can be prescribed to treat cataplexy and sleep paralysis. Several types of antidepressants are often used: selective serotonin reuptake inhibitors (SSRIs), serotonin and norepinephrine reuptake inhibitors (SNRIs), and tricyclic antidepressants (TCAs). SSRIs include fluoxetine, atomoxetine, and sertraline, SNRIs include Venlafaxine, and TCAs include protriptyline, clomipramine, and desipramine. Although SSRI may have fewer side effects, TCAs, SSRIs, and SNRIs have not been proved by clinical trials on the safety and efficacy in pediatric patients and are used off-label. 

Nocturnal sleep disturbance:

Nocturnal sleep disturbances of narcolepsy include sleep paralysis, hypnagogic/hypnapompic hallucination, and sleep fragmentation. Sodium oxybate is suggested for sleep paralysis and hypnagogic/hypnapompic hallucination for adult narcolepsy [60]. SSRIs, tricyclic antidepressants, and venlafaxine can be effective treatment for hypnagogic hallucinations and sleep paralysis as well [60]. However, there is no study available about the effectiveness and safety of these medications in patients of pediatric narcolepsy.

Medications under investigation:

There are also medications that may be effective for pediatric narcolepsy, but more studies are warranted for pediatric narcolepsy, including Mazindol, Pitolisant, and Solriamfetol.

Mazindol, a non-amphetamine, tricyclic compound, also have proved to be effective in patients presenting with narcolepsy and idiopathic hypersomnia. Current studies support use of mazindol in adult patients, but its safety and efficacy in pediatric patients has not been proved by clinical trials [73].

Recently, Pitolisant and Solriamfetol have been approved by the Food and Drug Administration to improve wakefulness in adults with narcolepsy. Pitolisant is a new wake-promoting agent. It is a selective histamine H3-receptor inverse agonist that acts presynaptically and can activate histamine neurons. For pediatric narcolepsy, a multicenter open-label, single-dose study of pitolisant 17.8 mg enrolled pediatric patients aged 6–17 years with narcolepsy. The exposure parameters were significantly greater in the younger compared with older patients and the doses up to 35.6 mg/d or 17.8 mg/d (body weight < 40 kg) are appropriate for pediatric patients. Six treatment-emergent adverse events were reported: headache (three), dizziness (one), vomiting (one), and diarrhea (one) [74].

Solriamfetol (JZP-110) is also a new high-potency wake-promoting agent, not yet approved for pediatric patients. It is a selective dopamine and norepinephrine-reuptake inhibitor and does not promote the release of monoamines. Its efficacy on both subjective and objective sleepiness was demonstrated in an international, double-blind, randomized, placebo-controlled trial recently, as well as its safety in both types of narcolepsy [75].

Immunomodulation strategy:

The development of immunomodulation therapy was based on the hypothesis of the autoimmune destruction of hypocretin neurons in narcolepsy, and the treatment may help to prevent such autoimmune reaction. Theoretically, in some patients, the treatment may slow down or decrease the destruction of orexin neurons. Although current studies show limited effects on reducing narcolepsy symptoms, studies have found that after receiving immunomodulation therapy, patients with more severe baseline narcoleptic symptoms achieved symptoms remission faster than controls [76,77]. 

Our clinical experience and concerns for pharmacological treatment:

In our clinical experiences, medications such as sodium oxybate were unavailable in Asian countries. Modafinil also had poor availability in most of Asian countries, except Taiwan and Republic of Korea. Hence, cranial nerve system stimulants such as methylphenidate still play a crucial role in treating pediatric narcolepsy patients in Asian countries. 

Generally, for treating EDS in pediatric narcolepsy, the desirable goals are to maintain monotherapy and avoid polypharmacy as possible [78], but in clinical practice, complex symptoms of narcolepsy, differences in the response to medications and comorbidities can make the combined use of different medications inevitable. 

Sodium oxybate can be prescribed as monotherapy to treat both EDS and cataplexy for pediatric narcolepsy patients [61,62]. Sodium oxybate can have extra treatment effects in night-time symptoms, but it is not available worldwide. Although there was no evidence to support the combination of multiple medications for pediatric narcolepsy, to enhance learning for older children or adolescents with increased academic stress, additional stimulants may be prescribed such as Methylphenidate. In some cases, the combined use of stimulants and antidepressants such as fluoxetine are needed for EDS and cataplexy. 

Children’s growth, including body weight and height should be monitored since the initiation of pharmacological treatment. Long-term use of medications in childhood narcolepsy may be limited due to adverse effects such as decreased growth. In some studies of pediatric patients with attention deficit hyperactivity disorder (ADHD), long term use of methylphenidate can result in reduction in height and weight [79]. In addition, there is evidence of body weight loss related to Modafinil in ADHD patients [80]. A 3 year study found body weight decreased after the first year treatment with sodium oxybate in pediatric narcolepsy patients [81].

Other side effects, such as hypersensitivity reactions of Modafini, and poor tolerability of TCA, were also reported [61,82]. To date, there are no RCT drug studies of Solriamfetol and Pitolisant in pediatric patients, and the efficacy and safety of these medications are still uncertain compared to currently approved medicines. 

Overall, the choice of medication should be individualized and depend on specific symptoms of each child. It may take weeks and/or trials of several different drugs to find out the appropriate medication and dosage with the least side effects.

2.Non-pharmacological approach:

Behavior modification should be provided to and discussed with children with narcolepsy and their caregivers, including:Regular sleep/wake schedule: Children with narcolepsy should have adequate nighttime sleep with a regular sleep/wake schedule, go to bed and wake up at the same time, and avoid staying up late.Good sleep environment: The bedroom should be quiet, dark, and cool, and the matrix should be comfortable. Avoid too much screen time before sleep. Children should avoid caffeine and stimulants before bedtime.Short naps: When the child feels most sleepy, a planned daytime nap is recommended. For example, short naps can be arranged daily, 3–4 times per day, about 10 to 20 min each time. The duration and frequency should base on each child’s needs.Exercise: Regular exercise can assist sleep and prevent obesity. Furthermore, children should eat meals at regular times and should not eat heavy meals or too much liquid close to bedtime.Relaxation before bedtime: Bedtime routines can include meditation, yoga, and music. A wake-up routine can also help to gain alertness in the morning, such as exercise in the sunlight or taking a shower.Risk prevention: We strongly suggest parents and teachers’ supervision when children with narcolepsy doing risky activities such as driving, swimming, or cooking. It is necessary for closely monitoring for safety concerns, and medication for wakefulness and cataplexy should be emphasized and prescribed according to individual needs.Education: Since narcolepsy is not common and well known, educating people around children with narcolepsy can be very important, including teachers, parents, family members, and close friends. They need to understand the disorder, how it affects functioning, and how they can assist the patients in dealing with it. Furthermore, teachers can also help to screen for narcolepsy if they possess a basic knowledge of the disorder, which can subsequently improve the delay in diagnosis. Psychoeducation group between teachers and students at school may also help the patients.Mental support: Children with narcolepsy face more difficulties and stress in their daily life. They are more likely to have depression or anxiety than children without the disorder. Therefore, support from mental health professionals or support groups can be helpful.

3.Management of comorbidities:

Depression and anxiety were the most common psychiatric comorbidities of pediatric narcolepsy [83]. A recent study found prevalence of depressive symptoms in patients with narcolepsy was 32%, with high between-study heterogeneity [84]. FDA approved antidepressants for depression and anxiety for children and adolescent such as fluoxetine can be used for symptoms control [85]. Suicide risk can increase with antidepressants [86], and thus monitoring and suicide prevention were essential.Attention deficit hyperactive disorder (ADHD) and oppositional defiant disorder (ODD) are also common comorbidities of pediatric narcolepsy patient [12,87,88]. A study with group of pediatric narcolepsy patients in Taiwan, found prevalence of ADHD and ODD were both 10.6% [12]. Methylphenidate is approved by FDA [89] and often prescribed in treating both daytime sleepiness and ADHD symptoms of children with narcolepsy. However, pediatric patients with narcolepsy have high levels of treatment-resistant attention-deficit/hyperactivity disorder (ADHD) symptoms and the response rate can be lower than children without narcolepsy [87]. The optimal treatment for ADHD symptoms in these patients warrants further evaluation in longitudinal intervention studies.The co-occurrence of psychosis in patients with pediatric narcolepsy was reported [90,91], but the correlation between narcolepsy and schizophrenia is still unknown [12]. Moreover, medication for narcolepsy including menthylphenidate can result in medication-induced psychotic symptoms [92], and there were also case reports with sodium oxybate and modafinil induced psychosis in adult narcolepsy patients [93,94,95,96]. Although research is less in pediatric patients, drug-induced psychosis has been reported in pediatric narcolepsy patients [97,98]. If psychosis is secondary to medication, tapering or even discontinuation of the suspected medication is recommended, and other medication can be chosen after the psychotic symptoms subside. For the comorbid primary psychosis such as schizophrenia, antipsychotic medications can be applied for the treatment of psychosis [91]. Aripiprazole is often prescribed first in our clinical practice, considering its less sedative side effect profiles. Other antipsychotic medications such as Brexpiprazole and Lurasidone were recently approved for prescription for adolescents, both of which also had less sedative side effect. However, pediatric patients with both schizophrenia and narcolepsy can have severe psychotic symptoms and can be difficult to treat in some cases [12]. Antipsychotic medications with stronger potency such as Risperidone and Olanzapine can be prescribed, although they may increase EDS.Narcolepsy patients exhibited high prevalence of obesity [97]. We suggest diet control and exercise as behavior modification and the compliance for pharmacological treatment of narcolepsy should emphasize to help maintain a more regular lifestyle and diet.Some studies found narcolepsy patients exhibited comorbidity with obstructive sleep apnea [98]. For treating OSA symptoms in pediatric narcolepsy patients, we suggest body weight control and may try continuous positive airway pressure (CPAP).

## 6. Conclusions

Narcolepsy is a life-long sleep disorder that currently has no cure. The diagnosis of narcolepsy in children can be difficult due to insufficient knowledge and under-recognition. The presentation of pediatric narcolepsy may differ from that of adult patients. EDS in children may manifest as irritability, hyperactivity, and poor attention, which may be incorrectly interpreted as misbehavior. Pediatric cataplexy can be subtle, such as unusual facial expressions, which are not observed in adults. 

Treatment of pediatric narcolepsy should involve the use of medication, behavior modification, education, and mental support. Only sodium oxybate had been proven effective in pediatric patients through randomized placebo-controlled studies. Other wake-promoting medications and antidepressants are used as off-label, and still warrant more studies to prove their efficacy and investigate possible side effects in pediatric narcolepsy. Although psychosocial interventions such as behavior modification is shown to benefit patients and the family, prospective long-term follow-up is necessary to evaluate the prognosis of outcome of children with narcolepsy.

## Figures and Tables

**Figure 1 children-09-00974-f001:**
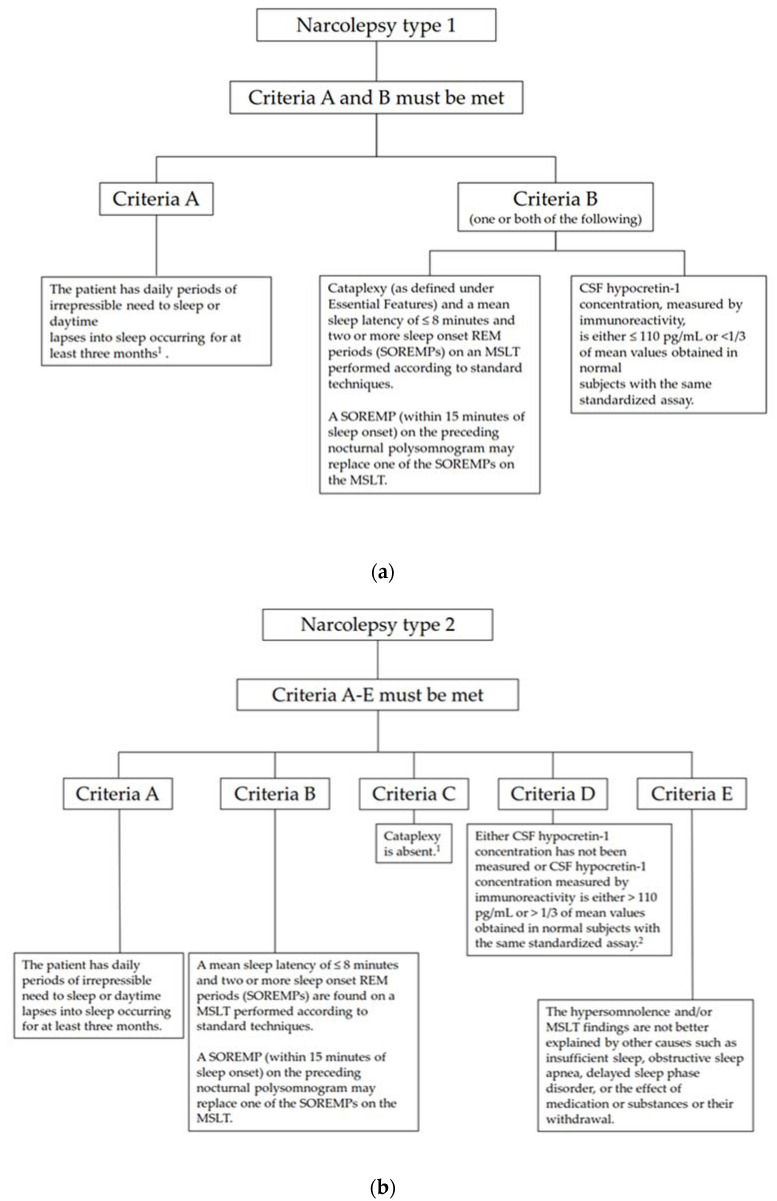
(**a**) Diagnostic criteria of narcolepsy type 1 according to International Classification of Sleep Disorders 3rd ed. [9]. In young children, narcolepsy may sometimes present as excessively long night sleep or as resumption of previously dis-continued daytime napping. Abbreviation: CSF: cerebral spinal fluid, MSLT: multiple sleep latency test, SOREMP: sleep onset rapid eye movement period. (**b**) Diagnostic criteria of narcolepsy type 2 according to International Classification of Sleep Disorders 3rd ed. [9]. If cataplexy develops later, then the disorder should be reclassified as narcolepsy type 1. If the CSF Hcrt-1 concentration is tested at a later stage and found to be either ≤110 pg/mL or <1/3 of mean values obtained in normal subjects with the same as-say, then the disorder should be reclassified as narcolepsy type 1. Abbreviation: CSF: cerebral spinal fluid, MSLT: multiple sleep latency test, SOREMP: sleep onset rapid eye movement period.

**Table 1 children-09-00974-t001:** Overview of medications available in pediatric narcolepsy.

	European Guideline Recommendation [57]	USA Guideline Recommendation [58]	FDA Approval (Narcolepsy Indication)	Level of Evidence [57]
Sodium oxybate	EDS Cataplexy DNS SP/HH	EDS Cataplexy	EDS (suggested) Cataplexy (suggested)	Low Low No data No data
Methylphenidate	EDS		EDS (suggested, children above 6 years old)	Very low
Modafinil	EDS Strong against patients with cataplexy	EDS	EDS (suggested)	Very low No data
Pitolisant	EDS *			Very low
Antidepressants	Cataplexy SP/HH			Very low Very low

EDS: excessive daytime sleepiness; DNS: disturbed nighttime sleep; SP: sleep paralysis; HH: hypnagogic/hypnopompic hallucinations. * with very limited data, in particular on safety, further evaluation is needed.

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
