# Peer review of "Pediatric Narcolepsy—A Practical Review"

_children, 2022, doi:10.3390/children9070974_

Round 1
Reviewer 1 Report
This review has been improved a lot. Just few comments as following:
1. The title is not consistent with the scope. Title seems only corresponding to treatment. Indeed, the scope is a review of narcolepsy given that so many details were given in each section.
2. Table 1 is not well organized. Concerning the diagnosis of narcolepsy, firstly, authors mentioned criteria A in both groups, respectively. Then the sequency is not in the same order. Please clarify them in the same order as: mean sleep latency, SOREMPs on MSLT for the diagnosis of narcolepsy; cataplexy and HCRT afterwards? Or using other clear way to present them. Criteria E is only suitable for NT2? Or for NT1 as well?
3. Please add references for “Prenatal nutrition, obesity, stress, and toxins have been factors suggested by some authors” on page 3, the last paragraph.
4. Given authors focus on the treatment on pediatric narcolepsy, it would be better to clarify the treatments in adults as well, further demonstrating the different treatment strategies for them and highlighting the treatment for children.
Author Response
Dear Editor,
Thank you and the reviewers for your precious comments, which assist us to improve the manuscript: children-1768468, “Therapies in Pediatric Narcolepsy – A Practical Review”. We have revised the manuscript according to your comments. To facilitate review, all text changes to the manuscript are marked up by ”Track Changes” function. The authors hope the quality of this revision is now appropriate for publication in Children.
Revisions in response to the reviewers’ comments are outlined below:
Respectfully,
Yu-Shu Huang
Dear Reviewer #1:
This review has been improved a lot. Just few comments as following:
Response: Thank you very much for your comments.
- The title is not consistent with the scope. Title seems only corresponding to treatment. Indeed, the scope is a review of narcolepsy given that so many details were given in each section.
Response: Thank you for your comments. Your reminder is important to us. We thoroughly revised our manuscript and add contents under another reviewer’s kindly suggestions. Moreover, according to our approach in treatment of pediatric narcolepsy, we wish to provide sufficient information for readers. In addition, early diagnosis is important for designing strategies in treating pediatric narcolepsy. Overall, we humbly offer comprehensive and practical information between sections, which nonetheless relate to our goal to present our review paper. We want to thank you again for your reminder, and we will revise the title into “Pediatric Narcolepsy – A Practical Review”.
- Table 1 is not well organized. Concerning the diagnosis of narcolepsy, firstly, authors mentioned criteria A in both groups, respectively. Then the sequency is not in the same order. Please clarify them in the same order as: mean sleep latency, SOREMPs on MSLT for the diagnosis of narcolepsy; cataplexy and HCRT afterwards?
Response: Thank you for your important comments. We review for diagnostic criteria from International Classification of Sleep Disorders 3rd edition. Under current criteria, the mean sleep latency, SOREMPs on MSLT, cataplexy, and HRCT all consists in differential narcolepsy type 1 and type 2. But the descriptions were separately mentions according to the criteria. For example, mean sleep latency, SOREMPs on MSLT, cataplexy, and HRCT all mentioned in criteria B from narcolepsy type 1, but were separately mentioned in criteria B, C and D of narcolepsy type 2. In order to present the differences comprehensively and clearly, we substitute the table with figure 1a and 1b.
Or using other clear way to present them.
Response : Thank you for your important comments. In order to present the differences comprehensively and clearly, we substitute the table with figure 1a and 1b.
Criteria E is only suitable for NT2? Or for NT1 as well?
Response : Thank you for your important comments. According to International Classification of Sleep Disorders 3rd edition, Criteria E was only mentioned in diagnostic criteria from narcolepsy type 2. We also presented the differences between type 1 and type 2 via figure 1a and 1b instead of original table (page 2).
- Please add references for “Prenatal nutrition, obesity, stress, and toxins have been factors suggested by some authors” on page 3, the last paragraph.
Response: Thank you for your suggestion. We added additional references with the relation between hypocretin and multiple factors. And we also revised the paragraph (page 4):
Some studies found factors such as prenatal nutrition, obesity and stress may also affect hypocretin neurons [26-28].
Reference:
- Iwasa T, Matsuzaki T, Yano K, et al. The effects of prenatal undernutrition and a high-fat postnatal diet on central and pe-ripheral orexigenic and anorexigenic factors in female rats. Endocr J. 2017;64(6):597-604.
- James MH, Campbell EJ, Dayas CV. Role of the Orexin/Hypocretin System in Stress-Related Psychiatric Disorders. Curr Top Behav Neurosci. 2017;33:197-219.
- Imperatore R, Palomba L, Cristino L. Role of Orexin-A in Hypertension and Obesity. Curr Hypertens Rep. 2017;19(4):34.
- Given authors focus on the treatment on pediatric narcolepsy, it would be better to clarify the treatments in adults as well, further demonstrating the different treatment strategies for them and highlighting the treatment for children.
Response: Thank you for your precious comments. According to your suggestion, we add a paragraph before the treatment section on page 5, to highlight the different aspects of treatment strategy between adult and pediatric narcolepsy patients:
Compared to adult narcolepsy patients, whose treatment strategy also included pharmacological and non-pharmacological approach, they had more approved medication for clinical use [58,59]. When prescribing medications for pediatric narcolepsy patients, it is important to acknowledge evident-based safety and the growth issues within each medication. It is also important to provide sufficient psychological counseling and behavioral modification for pediatric narcolepsy patients in multiple environments, and to enhance the cooperation between school teachers, family members and medical team.
Reviewer 2 Report
Thank you for possibility to review the manuscript titled Therapies in Pediatric Narcolepsy - A Practical Review.
The manuscript is well written, the flow is very good. Topic is of interest to both clinicians and researchers. I did not find any major flaws. The only one is that it is not PRISMA compliant and the references are not uniform and in accordance to the journal format. Several items are out-of-date (21, 45)
In summary, manuscript is excellent and worth publishing. The reading has been a real pleasure, congratulations!
Author Response
Dear Editor,
Thank you and the reviewers for your precious comments, which assist us to improve the manuscript: children-1768468, “Therapies in Pediatric Narcolepsy – A Practical Review”. We have revised the manuscript according to your comments. To facilitate review, all text changes to the manuscript are marked up by ”Track Changes” function. The authors hope the quality of this revision is now appropriate for publication in Children.
Revisions in response to the reviewers’ comments are outlined below:
Respectfully,
Yu-Shu Huang
Dear Reviewer #2:
The manuscript is well written, the flow is very good. Topic is of interest to both clinicians and researchers. I did not find any major flaws.
Response: Thank you very much for your comments.
The only one is that it is not PRISMA compliant and the references are not uniform and in accordance to the journal format.
Response: Thank you for your comments. We humbly presented a review article under the topic of “Pediatric Narcolepsy – A Practical Review”. With your kindly suggestion, we had adjusted and uniformed the references’ format.
Several items are out-of-date (21, 45)
Response: Thank you for your precious comments. We re-evaluated the importance from the reference:[21] Juji T, Satake M, Honda Y, Doi Y. HLA antigens in Japanese patients with narcolepsy. All the patients were DR2 positive. Tissue Antigens. 1984;24:316–9. In the section, we wished to present a historical aspect of relation between narcolepsy and human leukocyte antigen, which was essential for current research and diagnostic tool. As for the other reference from “Sleep 1994;17(8 Suppl):S60-S67”, we had replaced it with another article: Mignot E, Lin L, Rogers W, et al. Complex HLA-DR and -DQ interactions confer risk of narcolepsy-cataplexy in three ethnic groups. Am J Hum Genet. 2001;68(3):686-699.
In summary, manuscript is excellent and worth publishing. The reading has been a real pleasure, congratulations!
Response: Thank you very much for your comments.
Round 2
Reviewer 1 Report
Please verify each section is regarding childhood narcolepsy given the title is modified.